# A Utility Maximized Harvest Decision Model for Privately Owned Coniferous Forests in the Republic of Korea

**Hyunjin An and Sangmin Lee \*** 

Department of forest policy research, Korea Rural Economic Institute, Naju-si 58321, Jeollanam-do, Korea; hjan713@krei.re.kr
**\*** Correspondence: smlee@krei.re.kr; Tel.: +82-61-820-2193

**Abstract:** This study examined optimal harvesting decisions of multiple age-class private forests that maximize private forest owners' utility. For this analysis, we developed two scenarios. One scenario was to maintain the harvest level currently performed in the Republic of Korea (ROK) (baseline scenario), and the other was to harvest according to the harvest prescription derived from the discrete-time utility model with a multiple age-class forest (optimization scenario). For the baseline scenario, the cohort component approach was applied to predict changes in the forest's age structure under the given harvest level. For the optimization scenario, we applied a discrete-time utility model that can describe the consumption and cutting behavior of private forest owners who manage a multiple age-class forest. Then, we compared the changes of the timber supply level and forest structure dynamic by scenarios. The results showed that current harvesting in ROK is not at its optimal level. The baseline scenario results showed that if the current level of harvesting is maintained, a total of 1,315,000 m$^3$ of soft wood will be supplied annually. However, the average annual wood supply will increase to 11,522,000 m$^3$ under the maximized utility scenario. In terms of timber self-sufficiency, if all domestic wood produced is supplied as materials, the supply level from the optimization scenario will meet the government's policy goal of a 30% timber self-sufficiency rate. However, if the baseline scenario is maintained, supply shortages can be expected by 2050.

**Keywords:** forest stand management; optimal harvest decision; utility of forest owners; timber supply; multiple age-class forest

## 1. Introduction

The Republic of Korea (ROK) is a country with rich forests, with approximately 60% of the country's land cover classified as forestland. Although most of the country's forests were destroyed during the Korean War, the forests have recovered rapidly because of intensive planting and regeneration in 1970 under a governmental plan. As of 2020, the timber volume per hectare (ha) has increased by about 12 times compared to those of the 1960s. Despite abundant forest resources, only 15% of the total timber demand is supplied domestically and imported wood remains the primary raw material for timber products in the ROK. Although the private forest shares 68% of the total forest area, most private forest owners are not able to earn enough income by producing timber in the ROK.

The main reason for this is the government's policy of limiting harvest volumes. The forest protection law of the ROK controls the total amount of annual logging in the country. To harvest in private forests, a harvesting permit approved by the government is necessary and large clear cuts are strictly restricted. Regeneration after harvesting is legally stipulated, and the government subsidizes this cost. Strict penalties apply for logging, and forest land use changes, that lack permission.

Even though the total amount of harvesting in the ROK has gradually increased over the last decade, the harvest volume has been limited to around 2,000,000 m$^3$ annually since 2015. Average annual harvest levels during 2017–2018 were 2,325,000 m$^3$, and 97% of harvesting was in private forests [1]. Considering that the total projected demand of round wood in 2020 is about 15,000,000 m$^3$ [1], the domestic forest round wood supply is insufficient. A secondary reason is that the current unbalanced forest age structure might possibly hinder the sustainable timber supply. The bulk of the ROK's forest area (80%) is currently between 20 and 50 years old due to the three-decade planting program dating to 1970. Therefore, production will rise as major age classes mature and will fall after harvesting these major age classes. With these conditions, even though the area of forest under private ownership is 67% of the total forest area in the country, the private forests are not linked to the income of forest owners.

Most countries adopt one of two strategies when there is a lack of domestic material. They either simply import it or export to other countries where required from its own resources. However, in the case of raw wood materials, imports from overseas are expected to become more difficult. One reason is the wood supply deficit across the world, in Asia in particular [2]. As the middle class in Asian countries grows, the increasing demand for higher-value wood products, such as wood furniture and flooring, becomes more acute compared to in the past [2]. However, natural forests have been greatly declining and investment in higher-value wood plantations is insufficient [2]. The simultaneous effect of the increasing demand and declining supply will lead to an increase in the real price of raw wood materials [2]. A log export ban (LEB) policy in major timber exporting countries might accelerate the shortfall in the supply of raw wood material. The purpose of the LEB policy is to conserve a country's own forest cover and to induce local economic development [3]. Forest-rich countries, including Indonesia, the Russian Federation, and Malaysia, have implemented log export restrictions for a long time. Resosudarmo and Yusuf [3] reviewed economic and political impacts on LEB policy to examine Indonesian experience while implementing the policy in the 1980s and 2000s using the computable general equilibrium (CGE) model. They found that the LEB policy tends to reduce the volume of harvesting logs in the short run. The policy may eventually benefit the country in the long run, owing to the growing wood processing industries, but these industries may require more logs to be cut [3].

Difficulties in imports may call for more opportunities to increase the diverse use of domestic raw wood material because forest resources of the Republic of Korea (ROK) have been growing more than they are used. The government has recognized this problem and are currently pursuing new policy changes. Moreover, the Korean Forest Services proposed many programs to be consistent with promoting the use of domestic timber and timber products. This political goal has increased the supply of domestic timber from 15% to 30%. However, current factors including limited harvesting and unbalanced age structure are expected to hinder the achievement of 30% timber self-sufficiency rate policy goals. Several studies have suggested different approaches to the supply of wood from the domestic market. In early periods, the purpose of forest resource projections was supporting policy measures to prevent over-utilization, meanwhile guaranteeing a steady supply of raw material to the industry [4]. Recently, balancing wood production with other non-market values of forests, such as carbon sequestration and biodiversity, have become crucial [4]. Timber supply has traditionally been modeled using aggregate data, while individual harvest choices have been affected by forest capital stock conditions [5]. Polyakov et al. [5] developed aggregate timber supply models for four roundwood products in a seven-state region of the southern U.S. using a stand-level harvest choice model. The estimated elasticity values of softwood and hardwood sawtimber supply were 0.34 and 0.31, respectively. These results were consistent with previous studies, while the elasticity values of softwood and hardwood pulpwood supply were lower compared to the findings of previous studies. Detected cross-price elasticity between sawtimber and pulpwood supply indicated a dominant effect of sawtimber markets on pulpwood supply. Kilham et al. [6] parameterized an inventory-based business-as-usual (BAU) wood supply scenario in southwest Germany in the period between 2002

and 2012. Their research focused on a stratified method to predict the binary harvest decision. The harvest occurrence predicted by the stratified method was compared to the results of logistic regression, which was a commonly used approach to predict harvest decisions. The estimated results showed that each of the methods had strengths and weaknesses. However, considering the accuracy of projecting harvest proportions, the stratified method was more suitable to generate BAU timber supply scenarios.

In order for domestic timber to be used as a raw material for the timber industry, an effective harvesting plan should be established to mitigate supply sector risks. The general rule of thumb in forest management is to obtain sustainable production through appropriate harvest and planting decisions. Developing harvest decision-making tools for sustainable timber production is an important issue in the forestry field, and various previous studies have been examined. The oldest harvest strategy in forestry is maximum sustained yield, which implies forests should be harvested to maximize average growth. An alternative approach introduced by Faustmann [7] takes a financial perspective for the harvest strategy and suggests that forests should be harvested when the present value of net revenue is maximized. Linear programing has been widely used in forestry to select best flow and timing of timber harvesting. The Korea Forest Research Institute [8] predicted the amount of harvest volume that can achieve sustainable forest management principles using linear programing (LP), Fuzzy-LP, Geometric programing (GP), and Fuzzy-GP for Korean forests. The objective is to maximize harvest volume in the periods under the constraint of forest age structure and total areas. The results showed that the annual wood production potential varies according to the models, but that estimated production potential is approximately 7,000,000–8,000,000 m$^3$, which is about three times the current production level.

Linear programming (LP) is the primary economic analysis tool for public and private forest management planning. However, several conflicting goals should be incorporated into the model. This problem might be addressed when one goal is selected as the objective and the other as a constraint set [9]. Galatsidas et al. [10] suggested two LP models for the optimal design of the production and harvesting schedule of fuelwood produced from even-aged Oak forests. The first model aimed at sustainable fuelwood production in the context of area control. The second model achieved the maximization of the volume per unit of time and led to a steady-state forest. In the first LP model, aiming at sustainable fuelwood production, an initial unbalanced distribution of age class is being gradually transformed into a uniform distribution. At the end of the planning horizon, this forest would ensure a steady supply of fuelwood. However, the environmental drawbacks of early harvesting of young stands should be compensated by the economic benefits of supply security. However, initial irregularities in the age distribution are kept in perpetuity in the second LP model which aimed at maximizing the volume per unit of timber. This model is ecologically preferable because the stands are harvested at their maturity. Nevertheless, it could be economically unfavorable regarding securing the fuelwood supply due to great fluctuations in the market supply, which depends on the maturity of large share age classes. An et al. [11] examined forest management planning to enhance carbon sequestration using the LP model. Their study confirmed that the harvest prescription from LP derived the balanced age-class distribution in the national forests of Korea, but the solution did not achieve normal forests with the perfectly even-aged distribution. The harvest plan from LP would enhance yearly carbon sequestration in forests, and a shorter rotation age tends to call for more carbon sequestration and economic profit. However, their solution failed to suggest an optimal rotation age because the study only assumed single rotation management. Although a 50-year rotation generates the best performance in terms of both economic profit and carbon sequestration, it would be difficult to ensure that 50 years is the optimal rotation age under single rotation management.

Many studies have suggested various techniques for optimizing timber harvest and investment on public and private forest management. Johnson and Scheurman [12] presented a number of models that appear to be the basis for techniques commonly used to solve this optimization problem. They discussed the relationship of the Timber Resource Allocation Method (Timber RAM) to linear objective form,

and several iterative allowable techniques, including volume check, Simulating Intensively Managed Allowable Cut (SIMAC), and Short Run Allowable Cut (SORAC), are formulated as linear objective versions of the model. They also formulated the Economic Harvest Optimization (ECHO) model as a quadratic objective version. Abdullah et al. [13] developed an optimization model for timber harvesting planning based on the selective cutting technique. Their study subjected a maximum number of trees to be harvested and a minimum number of trees to be damaged during each planning period. Three optimization techniques were compared to solve the model: Monte Carlo programming, simulated annealing, and threshold acceptance. The results indicated that the simulated annealing method produced better solutions than the Monte Carlo programming and threshold acceptance. Sohngen et al. [14] developed a global timber market model that evaluated how timber supply reacts to future predicted increases for timber demand. The future demand for timber is expected to increase and it requires forest owners to increase future management intensity. The model predicted that rising timber prices would inspire new plantation and management intensity in the 21st century. Plantation will be expected to increase in warm and subtropical temperatures, but the boreal and tropical forests will be largely preserved due to low timber productivity and high access costs.

Another critical issue for forest management is incorporating environmental values such as carbon sequestration into harvest decision models. Forests are known as either carbon sources or carbon sinks [15]. Some carbon is lost when trees are harvested or when they decay. Standing trees absorb carbon from the atmosphere through photosynthesis. To properly assess the harvest decision, forest carbon should be incorporated since harvesting is considered a carbon-release source. Sohgen and Mendelsohn [16] developed an optimal control model of carbon sequestration and energy abatement to examine the potential role of forests in greenhouse gas (GHG) mitigation. They claimed that if GHGs accumulate in the atmosphere, the rental price of carbon sequestration should rise over time. The model showed that carbon sequestration is costly, but forest owners can sequester substantial amounts of carbon in forests mainly through increasing forestland and lengthening rotations. They claimed that carbon sequestration in forests is predicted to account for about one-third of total global carbon abatement. Mendelsohn and Sohngen [17] adopted a dynamic forest sequestration strategy that is consistent with the optimal control model of carbon mitigation in the ROK's forests. Their study investigated the potential of using the ROK's forests for harvesting and carbon sequestration. Although harvesting levels have been increasing over the last decade, the global timber model suggested that the harvest should be much higher as more stands reach maturity. They also suggested that forest carbon sequestration should be motivated by the carbon price of each decade

Since values of standing trees vary depending on their age class, age-class structure models have become more common in various forestry applications. Salo and Tahvonen [18] developed a discrete-time forestry model with nonlinear utility that can incorporate any number of forest age and land classes. The model process showed that multiple age-class forests showed a cyclical stationary state instead of an even flow of timber under discounting. The cyclical structure is dependent upon the initial age-class distribution and the cyclicality in total timber harvest vanishes as the discount factor approaches unity. However, the age-class structure may not approach a normal forest in any class. Generally, a significant part of the private forest owners' annual harvest decisions are still unexplained due to the difficulty of identifying the individual restriction of optimization behavior and forest owners' management motivation [19]. Therefore, the forest management model should recognize the diversity of forest owners' objectives. To take into account the multiplicity of objectives, theoretical models usually assumed that the forest owner obtains utility not only from income but also from the nonmarket amenity value of the forest [19]. Uusivuori and Kuuluvainen [20] examined the consumption and cutting strategy of a nonindustrial private landowner who was managing a multiple age-class forest using the discrete-time utility model. Their study assumed that forest owners would maximize their utility by consuming profits generated from their forests and other financial assets. The forest owners' consumptions are derived from both harvesting trees and the amenity provided by the standing trees. Their study introduces a decision variable matrix collecting the harvesting

shares of all the age classes, which would be suitable for defining a harvesting plan aimed at a specific forest structure. It also traces the dynamic path of the forest structure to the equilibrium forest. Uusivouri and Laturi [21] introduced a much simpler model to deal with harvest decision problems for multiple age-class forests. They derived the rules for optimal short- and long-run behavior of the landowners using an age structure model linked with the growth effect of the silvicultural investments. In their study, the utility-maximizing landowners faced optimization problems, which determined the optimal consumption path with two types of intertemporal management decisions, such as the optimal harvest age of even-aged stand and the amount of silvicultural investment for each stand.

This study has examined the optimal harvesting decision that is made for multiple age-class private forests using a discrete-time utility model, to help maximize the forests' utility. Investigating optimal operational techniques in different management environments is a central aspect of forestry research. A classic stand-level optimization problem in forest economics is the determination of the optimal harvest age for an even-aged stand to maximize the present value of an infinite series of timber generations and harvest cycles [7]. Even-aged stand management based on clear cutting has been the dominant strategy implemented across Europe since World War II [22]. Recent interest in uneven-aged management, based on selection cutting of individual trees or small groups of trees, has attracted a considerable amount of research which has utilized deterministic optimization [23]. Even though uneven-aged stand management strategies have been in development since the 1970s [24–27], their ecological and economic performance has been poorly examined as it requires a heavy computational burden [28]. For this reason, although large clear cutting is restricted in the ROK, studies for stand management decisions for ROK's forests are established assuming an even-age stand management strategy based on clear cutting. We believe however, that a multiple age-class management technique that would allow for selective cutting would be a more appropriate model for forests in the ROK. Moreover, evidence has shown that uneven-aged stand management decisions based on selection cutting adequately serves both economic and ecological goals [28].

This investigation hypothesized that the current harvesting level of privately owned coniferous forests in the ROK was not optimized, and this may partially explain the current inefficiencies of the timber sector. This study applied a modified discrete-time utility model with multiple age-class forests, as suggested previously by Uusivuori and Kuuluvainen [21], to help improve efficiency. Two harvest scenarios were established: the first maintained the ROK's current harvest method (status quo, baseline scenario), and the second utilized a harvest method derived from the discrete-time utility model (optimization scenario). The results compared the changes in the timber supply levels and the forest structure dynamics, using different scenarios to evaluate how the optimization scenario improved the supply sectors efficiency.

## 2. Materials and Methods

For the baseline scenario, the cohort component approach was applied. This approach is mainly applied to predict population growth, but this study used it to predict changes in the forests' age structure. For the optimization scenario, this study applied Uusivuori and Kuuluvainen's [21] discrete-time utility model that can describe the consumption and cutting behavior of private forest owners who manage a multiple age-class forest. This approach set out the utility-maximizing forest owners' optimization problem. The forest owner determines the optimal consumption path by management decisions such as determining the optimal harvesting age and volume of multiple aged forests. Uusivuori and Kuuluvainen's [21] study assumed that forest owners would maximize their utility by consuming profits generated from their forests and other financial assets. However, this study assumed that profits other than forestry are 0 for simplification. The profits from forestry include both selling timber derived from harvesting and amenity value derived from the standing trees. As with the general forest economic problem, this study also assumed that forest land generates the most utility, when it is used as forest. Therefore, the forest owners do not intend to use their forest

lands for other purposes by removing the trees, and the time range extends from zero to infinity. In other words, the forest land will remain as forest perpetually.

To evaluate whether each harvesting scenario could achieve the government's policy goal of a 30% timber self-sufficiency rate, the projected timber demand from An et al. [1] was used. By comparing the projected timber demand derived from An et al.'s model, we can predict the possible timber sufficiency rate under each scenario.

### 2.1. Harvest Scenario: Baseline

This baseline scenario seeks to examine the effects of an exogenous harvest plan which assumes that the total amount of logging is limited by a given level. To project the dynamics of forest age structure and harvest volume according to a given harvest level, this study developed the forest age-class (tree stands are classified into 6 age classes, each age class has a ten-year period. For example, trees aged from 1 to 10 years old are included in age class 1, and age class 2 includes trees aged from 11 to 20 years old) cohort component model. The cohort component relationship between forest age-class structure and time flowing can be expressed as the following equations. Assuming that there is no logging, the total area of $i$ age class for $t$-year ($t$-year implies time horizon) is the same as the total area of $i$-1 age class in $t$-1 to $t$-10, excluding the damaged areas due to forest fire, pests, and other factors. It can be expressed as the following equation:

$$A_t^i = \sum_{k=1}^{10} [(A_{t-k}^{i-1} - D_k^{i-1}) \times \frac{1}{10}] \tag{1}$$

where $A_t^i$ = Total area of age class $i$ in period of $t$,

$\quad A_{t-k}^{i-1}$ = Total area of age class $i$-1 in period of $t$-k,

$\quad D_k^{i-1}$ = Total area of tree death at age class of $i$-1 for $k$ years.

Assuming a certain tree mortality rate ($\varepsilon$) annually, the areas of dead trees can be expressed as follows:

$$D_k^{i-1} = \varepsilon(1-\varepsilon)^{k-1} A_{t-k}^{i-1} \tag{2}$$

where $A_t^i$ = Total area of age class $i$ in period of $t$,

$\quad \varepsilon$ = Mortality rate.

Substitute Equation (2) into Equation (1), to derive Equation (3):

$$A_t^i = \sum_{k=1}^{10} [(1-\varepsilon)^k A_{t-k}^{i-1} \times \frac{1}{10}] \tag{3}$$

Therefore, the forest area of the age class ($i$) in year $t$ would satisfy the cohort component relation following Equation (4):

$$\begin{aligned} A_t^i = &\sum_{k=1}^{10} (1-\varepsilon)^k I_{t-(k-1)}, \qquad i = 1 \\ &\sum_{k=1}^{10} [(1-\varepsilon)^k A_{t-k}^{i-1}) \times \tfrac{1}{10}], \quad i = 2,3,4,5 \\ &\sum_{i=6}^{n} \sum_{k=1}^{10} [(1-\varepsilon)^k A_{t-k}^{i-1}) \times \tfrac{1}{10}], \quad i = 6 \end{aligned} \tag{4}$$

where $I_{t-(k-1)}$ = Planting areas in year of $t-(k-1)$.

From Equation (4), the dynamics of each age class by year can be projected. The area of age class 1 (tree year 1–10) is estimated through the planted area for the next 10 years, and the harvested area in year $t-1$ is assumed to be the same as the planted area in $t$. In this scenario, the rotation age is also

determined by the government. Here, the age class *i* represents age class 1–6, and age class 6 includes the trees with age class 6 and older (age class n). This is because most forests in Korea have been generated since 1970, and only a few forests are older than age class 6.

## 2.2. Utility Maximized Harvest Scenario Considering Forest Carbon Sequestraion and Timber Value Scenario

### 2.2.1. Model and the Value of the Forest

A private forest owner is assumed to maximize the utility with his/her consumption level, and the consumption originates from profits generated from forestry and other assets. If the profit from other assets ($w$) is constant, the owner's consumption level will depend only on the forest profit, which is usually generated from timber and non-timber production. The forest profit can be expressed by the land value of the forest if the land is used only for forest, perpetually. This problem can be expressed as the following Equation (5). The owner of the forest will determine his/her consumption level and harvest rate to maximize utility.

$$\text{MAX}_{c_t, a_j^i} \sum_{t=0}^{\infty} \beta^t [u(c_t)]$$
$$\text{s.t.} \quad \sum_{t=0}^{\infty} \left(\frac{1}{1+r}\right)^t c_t \leq w + \sum_{i=0}^{n} LV(x^i) \tag{5}$$

As explained in the previous example, the utility is the concave function of the forest owner with logarithmic consumption. $\beta = \frac{1}{1+\rho}$ is the discount factor and $c_t$ implies consumption. $\rho$ is subjective preference rate that is greater than 0. $a_j^i$, which is the argument of land value equation (Equation (6) and (7)), is the harvest rate of trees at the age of *i* after *j* periods. $a_j^i$ can have a value between zero and one, inclusively, and the sum of it with respect to *j* is one. $x^i$ is the area of forest, which has trees at the age of *i*. The sum of $x^i$ is the total area of the forest. The constraint above Equation (5) is the lifetime budget condition: that the present value of consumption flow cannot exceed the initial value of assets, which is the forest land values, $LV(x^i)$, from age class zero (bare land) to age class n and other assets. Since the land is assumed to be used only for forest, the value of the land is the sum of all profits from the forest. If forest is used only for timber production, the value of the forest is the sum of the profits continually generated through the selling of timber and the costs incurred for timber production such as harvest and regeneration costs.

If the forest is filled with trees, there is no bare land. However, the forest after the harvest will become bare land. Therefore, the manager should know the land value without trees to evaluate the forested land value. The bare land value is determined by the present value of future income and cost flows related to the land. Thus, the timing and extent of management costs during a rotation and the timber selling profit and regeneration costs at the end of the rotation affect the bare land value [21]. Unlike entire initial bare land forest, the forest land is not usually tied to one single rotation cycle. The value of a forested area can be generated by the fixed bare land value plus the value of cash flow related to the standing timber volume and its growth. Uusivuori and Kuuluvainen [21] showed that if forests are started in bare land, and if given parameters such as discount rates and timber prices are constant, the harvest plan approaches a long-term steady state where consumption and timber volume are fixed at a certain level. However, within a forested area, the optimal harvest plan may differ from the case of starting with bare land. Since the non-timber value that occurred prior to the harvest plan does not affect the optimal rotation age decision, the derived optimal harvest plan would reach a short-term equilibrium in the case of forested areas.

Assuming total profit from forests is the sum of profit from carbon sequestration in standing trees and harvesting mature timbers, the bare land value can be expressed as Equation (6) below. Here, *k* denotes the cost of harvesting and replanting, $P_t$ denotes the wood price, and $q^0_{jj}$ denotes the volume ($m^3$/ha) per unit area of forest starting from bare land and growing to age *j*. The first *j* of subscription indicates the harvest period and the second one describes the duration of periods after

afforestation or reforestation of bare land. Also, $r$ is the discount rate, $P_{CO2}$ is the market price of $CO_2$, $\delta$ is the $CO_2$ conversion factor per unit timber volume (ton/m$^3$), $T_h$ is the forest management cost, and $x^0$ is the bare land area (ha).

$$LV(x^0) = -kx^0 + \frac{1}{1+r}(P_t q^0{}_1 - k)\frac{1+r}{r}a^0{}_1 x^0$$

$$+ \sum_{j=2}^{\infty}\left[\left(\frac{1}{1+r}\right)^j(P_t q^0{}_{jj} - k) + \sum_{h=1}^{j-1}\left(\left(\left(\frac{1}{1+r}\right)^h - \left(\frac{1}{1+r}\right)^j\right)P_{CO2}\delta\Delta q^0{}_{jh} - \left(\frac{1}{1+r}\right)^h T_h\right)\right]$$

$$\times \frac{(1+r)^j}{(1+r)^j - 1}a^0{}_j\prod_{h=1}^{j-1}(1 - a^0{}_h)x^0$$

(6)

To simplify the problem, it is assumed that trees do not absorb $CO_2$ during the period just before the harvest. We also assume that about 50% of $CO_2$ sequestered in the tree is released upon cutting [9]. Therefore, the forest owner's profit from $CO_2$ sequestration could be the present value of the revenue generated through $CO_2$ sequestration during the period of tree growth minus the future cost of carbon emission through logging. Harvesting is assumed to be performed at the end of one period, and one period is 10 years. When trees are planted in a given bare land area $x^0$ and the harvesting rate for $x^0$ after a period is $a^0{}_1$, the corresponding forest area in the next period is $(1-a^0{}_1)\,x^0$. If harvesting is continued in this way, the harvesting area of period $j$ is $a^0{}_j\prod_{h=1}^{j-1}(1 - a^0_h)x^0$.

To calculate the value of forest land, which already has trees on it, the unit value of the bare land is used as in Equation (7). The value of forestland of age $i$ will be defined by the discounted profit generated from it until the time of harvest plus the fixed bare land value $LV(x^0)$ of the harvest area.

$$LV(x^i) = \left[P_t q^i{}_{00} + \frac{LV(x^0)}{x^0}\right]a^i{}_0 x^i$$

$$+ \sum_{j=1}^{\infty}\left[\left(\frac{1}{1+r}\right)^j\left(P_t q^i{}_{jj} + \frac{LV(x^0)}{x^0}\right) + \sum_{h=1}^{j-1}\left(\left(\left(\frac{1}{1+r}\right)^h - \left(\frac{1}{1+r}\right)^j\right)P_{CO2}\delta\Delta q^i{}_{jh} - \left(\frac{1}{1+r}\right)^h T_h\right)\right]$$

(7)

$$\times a^i{}_j\prod_{h=0}^{j-1}(1 - a^i{}_h)x^i$$

Here, similar to $q^0{}_{jj}$ in Equation (6), $q^i{}_{jj}$ denotes the timber volume (m$^3$/ha) per unit area of forest starting from age class $i$ and growing to age $i + j$. If the volume is harvested after $j$ periods, the harvest rotation becomes $i + j$. Accordingly, $q^i{}_{jh}$ represents the volume of per unit area of trees in age $i + h$ that started with age $i$ and have $i + j$ rotation age. Therefore, in this case, the trees were in age $i$ in the beginning of the forest planning. They remain uncut for the period $j$ and the forest owner obtains profits from the standing trees.

### 2.2.2. The Optimal Harvest Strategy for Multiple Age-Class Forested Area

The optimization problem of forest owners can be investigated using the forest value. The following first-order condition can be derived by the Lagrange equation. Here, $\lambda$ is a Lagrange multiplier [21], which represents the present value of marginal utility of consumption. The condition implies that consumption will be an increasing plan when the subjective time preference ($\rho$) is lower than the discount rate ($r$) and vice versa. If the subjective time preference ($\rho$) is equal to the discount rate ($r$), the consumption plan is constant.

$$\frac{\lambda}{u'(c_t)} = \left(\frac{1+r}{1+\rho}\right)^t$$

(8)

Another necessary condition is to differentiate the Lagrangian function with respect to $a^0{}_j$ and $a^i{}_j$. Through this process, we can compare the profit generated from harvesting by each age class and the profit generated from delaying harvest one period. For the optimal harvesting rule, we should determine the rotation length, which is the length of time a tree will grow before it is cut. If the

forest is grown from bare land, the optimal harvesting rule will give the optimal rotation that is followed in perpetuity after the first rotation. In the case of forested areas, an optimal rule gives the number of periods which they are left to grow before the first clear cutting. If the forest is started from bare land, we can derive the optimal rotation using Formula (9). The basic concept of Formula (9) follows the Faustmann rule that compares the present values of marginal timber profits obtained by following rotation $j$ and $j + 1$ with the value generated from $CO_2$ sequestration in standing trees (the partial derivative of Lagrangean function with respect to $a^0_j$ results in the marginal bare land value of harvest at age j, which is the partial derivative of bare land value (Equation (6)) with respect to $a^0_j$. This value would be compared with marginal bare land value of harvest at age $j + 1$, which is the partial derivative of Equation (6) with respect to $a^0_{j+1}$). If the profits from harvesting, both timber and carbon sequestration, in tree age $j$ are greater than the profits from delaying harvest in tree age $j + 1$, then harvesting is the better decision. In this case, $a^0_j$ becomes 1. In the opposite case, delaying harvesting is the better decision. If the values on both sides are the same, part of the land is allocated to forestry following rotation $j$ while another part of the land is allocated to forestry following rotation $j + 1$ [21]. The harvest rate ($a^0_j$) can have some values between zero and one, if the values of right-hand side (RHS) and left-hand side (LHS) of Equation (9) are equal. However, in a discrete-time problem, those values cannot be the same, so the value of harvest rate is always zero or one.

$$
\begin{aligned}
&[(\tfrac{1}{1+r})^j (P_t q^0_{jj} - k) + \sum_{h=1}^{j-1} (((\tfrac{1}{1+r})^h - (\tfrac{1}{1+r})^j) P_{CO_2} \delta \Delta q^0{}_{jh} - (\tfrac{1}{1+r})^h T_h)] \tfrac{(1+r)^j}{(1+r)^j - 1} \\
&> [(\tfrac{1}{1+r})^{j+1} (P_t q^0_{j+1,j+1} - k) + \sum_{h=1}^{j} (((\tfrac{1}{1+r})^h - (\tfrac{1}{1+r})^{j+1}) P_{CO_2} \delta \Delta q^0{}_{j+1,h} - (\tfrac{1}{1+r})^h T_h)] \tfrac{(1+r)^{j+1}}{(1+r)^{j+1} - 1}
\end{aligned}
\tag{9}
$$

Once the optimal rotation for bare land is determined, the optimal rules for the forested area can be derived using Formula (10), where $LV(x^0)$ represents the bare land value. The condition of Formula (10) implies that the tree should be harvested at the age of $j$ if the marginal benefit of harvesting at the age of $j$ is greater than the marginal benefit of harvesting at $j + 1$ (in this case, the marginal land value of harvest at age $j$ would be compared with marginal land value of harvest at age $j + 1$, which are partial derivatives of Equation (7) with respect to $a^0_j$ and $a^0_{j+1}$, respectively). If the inequality sign is opposite, then delaying harvest for one period is the better choice. This rule can be applied to each stand.

$$
\begin{aligned}
&[(\tfrac{1}{1+r})^j (P_t q^i_{jj} + \tfrac{LV(x^0)}{x^0}) + \sum_{h=0}^{j-1} (((\tfrac{1}{1+r})^h - (\tfrac{1}{1+r})^j) P_{CO_2} \delta \Delta q^i{}_{jh} - (\tfrac{1}{1+r})^h T_h)] \\
&> [(\tfrac{1}{1+r})^{j+1} (P_t q^i_{j+1,j+1} + \tfrac{LV(x^0)}{x^0}) + \sum_{h=0}^{j} (((\tfrac{1}{1+r})^h - (\tfrac{1}{1+r})^{j+1}) P_{CO_2} \delta \Delta q^i{}_{j+1,h} - (\tfrac{1}{1+r})^h T_h)]
\end{aligned}
\tag{10}
$$

The formula for calculating a forest's value (Equations (6)–(7)) and the formula for harvest determination (Formulas (9) and (10)) are used to analyze the optimal harvest strategy for privately owned coniferous forests in the ROK. The first step needed for analysis is to determine the rotation length for the bare land using Formula (9). Using the given parameters and the volume per ha, the profit from harvesting by age class can be calculated to determine the optimal harvest decision. Since the model assumes a discontinuous timeline, the results of the harvesting plan can be shown by the following two forms: to harvest all trees in a certain age class ($a = 1$) or to delay harvesting ($a = 0$) of all the trees in a certain age class. Once the rotation length for bare land is determined, the value of bare land can be calculated using Equation (6). Then, the rotation length for a forested area can be determined if we substitute the bare land value into Formula (10). Putting all this together, the value of forested areas can be calculated using Formula (10). The value of forested area derived through this process is equal to the permanent consumption of private forest owners, assuming that the value of non-forestry assets is zero. Therefore, the amount of annual consumption can be obtained by substituting the derived value of a forested area into Equation (8), since $\lambda = (\tfrac{1+r}{1+\rho})^t \cdot u'(t)$. Since the

model assumes the discount rate ($r$) is equal to the subjective time preference ($\rho$), the consumption plan has a constant value regardless of time. The amount of annual consumption is equal to the amount of annual harvesting in monetary terms if the non-forestry asset is zero. Dividing this value by timber price yields the optimal harvesting volume.

## 2.3. Data and Coefficients

For the analysis, the model was applied to a privately owned coniferous forest in the ROK because the species used for timber in the ROK are mainly conifers. Non-coniferous trees are used for wood chip or pulp and are rarely used as timber. Data for the country's entire area of coniferous forest by age class and property can be obtained through the Statistical Yearbook of Forestry published by the Korean Forest Service [29]. The timber volume per unit area by age class for coniferous forests can be calculated using data from the National Institute of Forest Science. We obtained the data of timber volume by age class for coniferous forest in Korea, then calculated weight average using the species distribution ratio. The site index applied to each species is the following: Gangwon Local Pine (*Pinus densiflora for.erecta*): 14, Red Pine (*Pinus densiflora* Siebold & Zucc): 12, Rigida Pine (*Pinus rigida mill*): 14, Korean Pine (*Pinus koraiensis*): 14, Japanese Larch (*Larix kaempferi*): 18, Hinoki cypress (*Chamaecyparis obtuse*): 12, etc. [30] (Table 1). For convenience, in both scenarios, 2015 was set as the start year (period 0), and each period is defined as 10 years.

**Table 1.** Areas and volume of coniferous forest in Korea: by age class.

|  | Age Class 1 | Age Class 2 | Age Class 3 | Age Class 4 | Age Class 5 | Age Class 6 |
|---|---|---|---|---|---|---|
| Area | 73 | 45 | 308 | 908 | 348 | 49 |
| Volume per ha | 36 | 91 | 163 | 214 | 252 | 279 |

Unit: 1000 ha, 1000 m$^3$.

Since most of the forests in the ROK were generated in the 1970s to recover forestland destroyed by the war, fast-growing species such as red pine (*Pinus densiflora*) were used and now account for about 50% of all coniferous trees in the country. For this reason, the official forest statistics only provide the overall statistic of coniferous forest, and data is not displayed by single species categories. However, other coniferous species besides red pine are not expected to account for a high proportion. Although the model used the overall coniferous forest data across the country, due to data limitations, we accounted for the differences in species indirectly through the timber volume per unit area. We calculated weight average using a species distribution ratio. The partial derivative of Lagrangian function with respect to $a^0_j$ results in the marginal bare land value of harvest at age j, which is the partial derivative of bare land value (Equation (6)) with respect to $a^0_j$. This value would be compared with marginal bare land value of harvest at age $j + 1$, which is the partial derivative of Equation (6) with respect to $a^0_{j+1}$. For site index, we applied an average site index value for each species.

The following coefficients are used for the model. $CO_2$ price is assumed to be 20 euros per 1 $CO_2$t based on the 2018 market price of carbon credits [31]. The carbon reserve in standing trees was calculated using the following equation [32]:

$$C_t = V \times D \times BEF \times CF$$

where $C_t$ is carbon reserved in a standing tree measured in tons, V is volume of trunk, D is average trunk density, BEF is biomass expansion factor (ratio of above ground biomass to trunk biomass), and CF is forest carbon content (0.5), ton of carbon/ton of dry matter [32]. The values of the above coefficients for coniferous forests were obtained from research by the National Institute of Forest Science [33]. For data on wood prices and costs, refer to Reference [34], and weighted average with the distribution ratio of each species (Table 2). For the subjective time preference ($\rho$), we assume $\rho$ is equal to the interest rate (r).

**Table 2.** Areas and volume of coniferous forest in Korea: by age class.

| Coefficient | Symbol | Value | Source |
|---|---|---|---|
| Discount Rate | r | 0.025 | |
| Price of $CO_2$ (Euro) | $P_0$ | 20 | [31] |
| Korean Won/Euro exchange rate | | 1.296 | August 2018 |
| $CO_2$ conversion factor (ton/m$^3$) | $\delta$ | 1.54 | [33] |
| Price of Conifer wood (1000 Won/m$^3$) | $P_t$ | 104.2 | [34] |
| Harvesting and regeneration cost (1000 Won/ha) | k | 11,178 | [34] |

## 3. Results

### 3.1. Baseline Scenario

Table 3 below shows the annual harvesting volume of national and private forests reported by the Korean Forest Service. The rotation age for the privately owned coniferous forest assigned by the Korean Forest Service is 45 years (Korean Forest Service specifies the rotation age by species and ownership according to government guidelines) (age class 5). To obtain the future harvest level for the baseline scenario, we averaged out harvest volume by ownership and species in 2017 and 2018. The baseline scenario assumes that forests are harvested at the same level in the future. In other words, annual harvesting volume from private coniferous forests is about 1,315,000 m$^3$, and trees will be harvested at this same level in the future. Then, the harvesting area is calculated by dividing the amount of harvest volume by volume per area (ha) of the species and age class.

**Table 3.** Annual harvesting performance: by species and ownership.

| | 2014 | 2017 | 2018 | 2017–2018 Average | Rotation Age | Volume per ha |
|---|---|---|---|---|---|---|
| Total | 1724 | 2291 | 2359 | 2325 | - | - |
| Coniferous Forest: Private | 997 | 1329 | 1301 | 1315 | 45 years | 175 m$^3$ |
| Coniferous Forest: National | 35 | 47 | 70 | 56 | 60 years | 275 m$^3$ |
| Broadleaf forest: Private | 677 | 904 | 958 | 931 | 35 years | 120 m$^3$ |
| Broadleaf forest: National | 14 | 13 | 30 | 22 | 40 years | 195 m$^3$ |

Source: Korea Forest Service. Available online: http://www.forest.go.kr (assessed on 3 July 2019). Unit: 1000 m$^3$.

Figure 1 shows changes in the area of future age-class distribution under the baseline scenario. Currently, the area of the forest is not evenly distributed across all age classes, and about 80% of the area is between age class 2 and 5. However, as time passes, the proportion of the young tress such as age classes 1 and 2 are expected to decrease while the area of age class 6 (here, the age class 6 includes the trees with age class 6 and older trees) is expected to increase significantly.

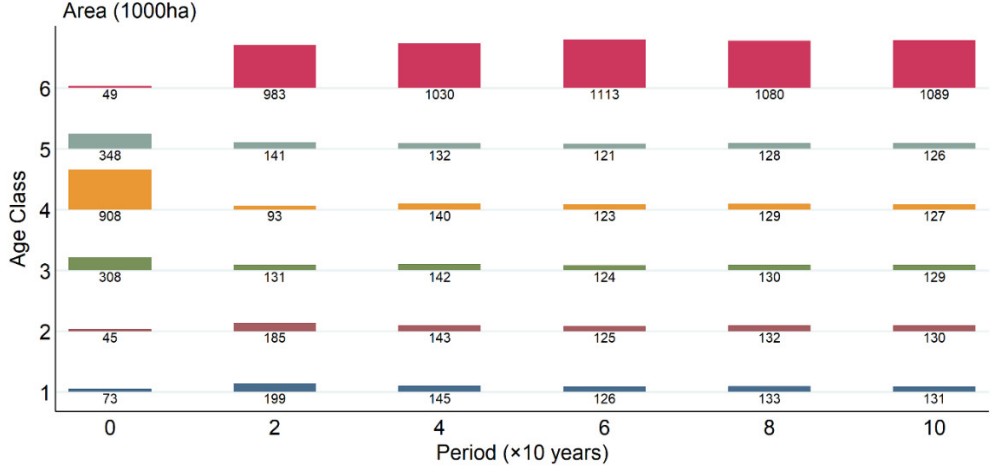

**Figure 1.** Changes in forest area by age class: Private coniferous forest.

*3.2. Utility Maximized Harvest Scenario*

As shown in Table 4, the optimal rotation length for forest started in bare land is age class 4, since harvesting at age class 4 generates the largest profit (LHS). The calculated result for the bare land value of privately owned coniferous forest in Korea, using Equation (6), is estimated at 392 billion Won, which is 226,300 Won per ha. To evaluate the optimal harvest rule for the forested area, we applied Formula (10). The result shows that postponing harvesting (RHS) would generate decreasing profit if the age of standing growth is between age classes 4, 5, and 6 (Table 5). The harvest strategy shows that cutting trees in age class 4, 5, and 6 at the same time is optimal. The forest owner, however, will be faced with the constraint (Equation (5)) that limits the periodic consumption and harvest volume, which depends on the land value. Consequently, it is required to find the maximum land value that will lead to the highest periodic consumption level by allocating periodically allowed harvest area between age class 4, 5, and 6.

**Table 4.** Profits generated from harvesting by age class: forest started in bare land.

|  | **Age Class 3** | **Age Class 4** | **Age Class 5** | **Age Class 6** |
|---|---|---|---|---|
| LHS (left-hand side of Formula (9)) | 2970 | 11,404 | 11,200 | 10,474 |
| RHS (right-hand side Formula (9)) | 11,404 | 11,200 | 10,474 | - |

Unit: 1000 Won/ha.

**Table 5.** Profits generated from harvesting by age class: forested area.

|  | **Age Class 3** | **Age Class 4** | **Age Class 5** | **Age Class 6** |
|---|---|---|---|---|
| LHS (left-hand side of Formula (10)) | 9880 | 22,553 | 26,429 | 29,340 |
| RHS (right-hand side of Formula (10)) | 18,426 | 21,227 | 23,356 | 24,670 |
| RHS–LHS | 8546 | −1326 | −3073 | −4670 |

Unit: 1000 Won/ha.

To evaluate the value of a forested area, two harvesting decisions are considered based on the result shown in Table 5: (1) preferential harvesting of age class 6 since it generates the greatest decreasing in profits due to postponing harvesting, and (2) simultaneous harvesting of the age classes 4–6. Table 6 shows the estimated value of a forested area according to each harvesting decision. Then, Equation (8) is used to evaluate the optimal harvesting volume for each harvesting decision. In the case of preferentially harvesting age class 6, the amount of harvesting volume is 115,217,000 m$^3$ over 10 years. If we choose simultaneous harvesting of age class between 4, 5, and 6, the amount of harvesting volume is about 102,800,000 m$^3$ over 10 years (Table 7).

**Table 6.** Estimated value of forested area by age class, by harvesting decision.

|  | Total | Age Class 1 | Age Class 2 | Age Class 3 | Age Class 4 | Age Class 5 | Age Class 6 |
|---|---|---|---|---|---|---|---|
| Preferential Harvesting of age class 6 | 54,850.2 | 1200.4 | 1026.8 | 9522.9 | 28,935.1 | 12,244.0 | 1921.0 |
| Simultaneous harvesting of age class 4, 5, 6 | 48,919.3 | 947.4 | 830.8 | 8041.2 | 26,202.6 | 10,976.3 | 1921.0 |

Unit: 10 billion Won.

**Table 7.** Amount of harvesting volume harvesting decision.

|  | **Amount of Consumption over 10 Years** | **Amount of Harvesting Volume over 10 Years** |
|---|---|---|
| Preferential Harvesting of age class 6 | 12,001.3 | 115,217 |
| Simultaneous harvesting of age class 4, 5, 6 | 10,703.6 | 102,758 |

Unit: 10 billion Won, 1000 m$^3$.

Summarizing the above results, preferential harvesting of age class 6 is a better decision in terms of profit and timber supply than simultaneous harvesting of age class 4–6 for privately owned coniferous forests in the ROK.

The area of each age-class area at the beginning of the management plan is provided, and the harvest rate from our analysis is zero or one. These results suggest that the area of each age class would be constant if the consumption level is the same as the profit from the forest area of harvest age in each period. However, the area of each age class changes until it approaches a constant level because much larger profit generated from the area of harvest age class than the manager's consumption level allows the harvest of a partial area from the age class. Figure 2 shows the changes of age-class structure over time under the strategy of preferentially harvesting age class 6. In this strategy, age class 6 would be harvested first, but the amount of harvesting volume of age class 6 cannot satisfy the optimal amount of harvesting volume (115,217,000 m$^3$ over 10 years) that maximized the forest owner's utility, and so the following lower ages such as age class 4 and 5 are assumed to be harvested. For the beginning of the period, the harvest volume from age classes 6 and 5 can satisfy the amount of optimal harvesting volume. In period 4, trees in age class 4 can begin to be harvested, as this is the optimal rotation age. According to the harvest prescription derived from our model, the harvesting area of age class 6 would be gradually decreased in period 4.

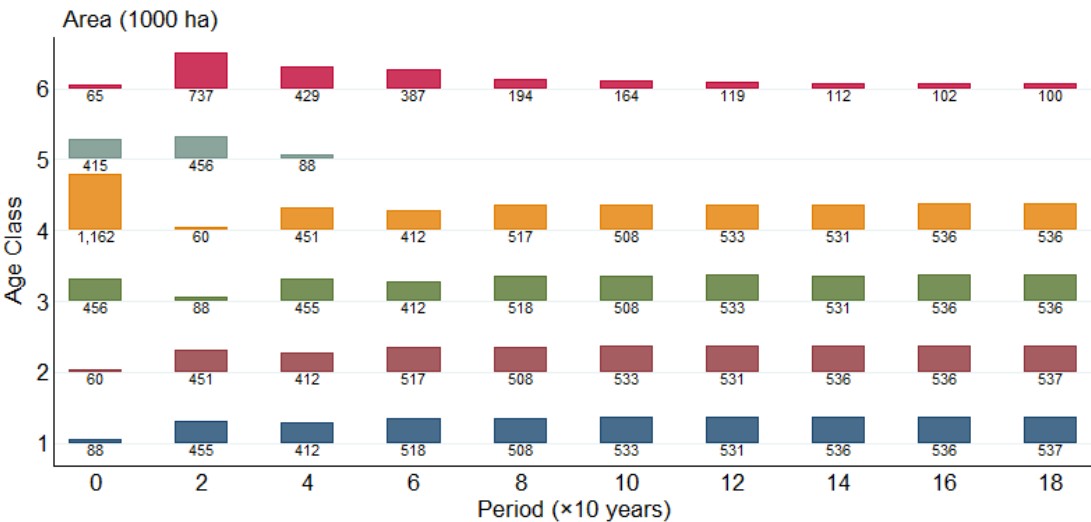

**Figure 2.** Changes in forest area by age class.

The harvesting area is about 451 thousand ha at the begging of the period when age class 6 would be preferentially harvested. However, from the second half of the planning period, most of the harvesting was performed in age class 4. The harvesting area tends to increase as time passes, since the tree volume of age class 4 is relatively small compared with age class 6. The harvesting area will be about 536,000 ha by period 18.

Figure 3 shows the changes in volume of forest stock and harvest volume over time when following the prescription harvest we derived. The result shows that the overall forest stock is gradually decreased while maintaining a constant harvesting volume. The reason is that as time goes by, the area of age classes 5 and 6 that occupy a high volume of forest stock per unit area is decreased, but the area of young trees in age classes 1 and 2 is increased. While the harvesting volume is maintained at a constant level of 115,217,000 m$^3$, the volume of forest stock is expected to be gradually decreased from about 454,600,000 m$^3$ at the beginning of the period, to 298,300,000 m$^3$ by period 18 (180 years later).

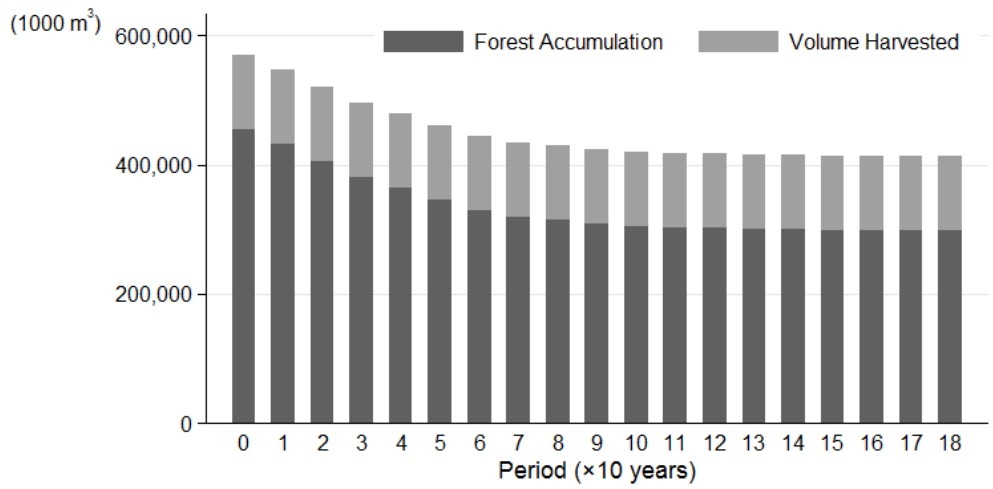

**Figure 3.** Changes in forest accumulation and volume harvested.

To summarize the harvest plan based on the above analysis results, the amount of harvesting that maximizes the utility of forest owners would be 11,521,700 m$^3$ per year. This implies that many more trees should be harvested than the actual current level of harvesting. In the short-term, which refers to current forested area, age class 6 would be harvested preferentially, but in the long-term, which refers to the replanted forests after harvesting current forest, age class 4 is then determined to be the optimal harvest target.

### 3.3. Timber Self-Sufficiency and Forest Owners' Utility

To examine if the supply derived from these two harvesting scenarios can meet the domestic demand of round wood, the projection results from An et al. [1] are used. They projected the future supply for wood products in the ROK and then estimated the demand of round wood. The wood product volume is converted into round wood volume using the round wood equivalent volume index. The calculated demand of round wood for the production of wood product is as follows in Table 8 [1]. The estimated results show that the demand for domestic round wood is expected to increase, while the demand for imported round wood is expected to decrease. However, the total demand for round wood to produce wood products is expected to decrease due to population decline and softening market demands.

**Table 8.** Projected demand of round wood to produce timber products.

| Year | Demand of Domestic Round Wood | | | | Demand of Imported Round Wood | | |
|---|---|---|---|---|---|---|---|
| | **Total** | **Sub-Total** | **Soft Wood** | **Hard Wood** | **Sub-Total** | **Soft Wood** | **Hard Wood** |
| 2020 | 15,005 | 2579 | 1882 | 697 | 12,426 | 10,374 | 2052 |
| 2030 | 14,760 | 2764 | 2035 | 730 | 11,996 | 9872 | 2124 |
| 2040 | 14,482 | 2905 | 2165 | 739 | 11,577 | 9442 | 2135 |
| 2050 | 14,236 | 3041 | 2295 | 745 | 11,195 | 9060 | 2135 |
| 2060 | 14,004 | 3174 | 2425 | 749 | 10,830 | 8699 | 2131 |

Source: An et al [1], p. 89. Unit: 1000 m$^3$.

As shown in Table 9, the projected demand of coniferous round wood (soft wood) in 2050 is 11,355,000 m$^3$. To achieve the policy goal of a 30% self-sufficiency rate in timber, at least 3,407,000 m$^3$ of domestic round wood is required per year. The projected timber supply under the utility maximized scenario would be sufficient to achieve this policy goal. However, timber supply is expected to be insufficient by 2050 under the baseline scenario (Table 9). Not only does the baseline scenario fail

to achieve the policy goal, it is expected meet only about half of the projected demand of domestic coniferous round wood.

**Table 9.** Projected timber demand and amount of available supply by scenarios.

| | Scenario | Total | Domestic Timber | Imported Timber | Required Domestic Timber for 30% Self-Sufficiency Rate |
|---|---|---|---|---|---|
| | Projected Supply | 11,355 | 2295 | 9060 | 3407 |
| Amount of | Baseline | 1375 | 1375 | - | 2032 |
| Available supply | Utility maximized | 11,522 | 11,522 | - | 3407 |

Unit: 1000 $m^3$.

Since the amount of annual consumption equals the amount of annual harvesting in monetary terms, the utility level of the forest owners can be derived by substituting this value into the utility function $u$ (·) in Equation (5). $u$ (·) is a concave function that represents the function of utility for a consumption level. Since $u$ (·) is a concave function with logarithmic form [21], the utility of forest owners over 10 years by each scenario is shown in Table 10. If we follow the harvest decision in accordance with the utility maximized scenario (B), the utility level of forest owners would be increased by 15% over 10 years compared with harvesting decisions in accordance with the baseline scenario. Utility is a relative concept, and the rate of change is more significant than the absolute difference in the value. The result confirms that if the harvesting level is higher than that of scenario B, the improvement in the utility is not significant since scenario B shows the optimal level of harvesting.

**Table 10.** Utility of forest owners over 10 years by scenarios.

| Scenario | Consumption over 10 Year | Utility over 10 Year | Changes in Utility (A vs. B) |
|---|---|---|---|
| Baseline scenario (A) | 1,369,744 | 14.13 | - |
| Utility Maximized scenario (B) | 12,001,307 | 16.30 | +15% |

## 4. Discussion

As the trees planted in the 1970s have matured, full-scale timber production is necessary to make valuable use of resources to increase income of forestry workers, improve forest structure, and to replant forests with species of high economic value. Although it is an abundant forest resource, the efficiency of the ROK's domestic wood supply has been continuously questioned since the forests are not linked to the income of forest owners. The analysis in this study shows that the current harvest yield is not at an optimal level, and it is not sufficient to reach the goal of achieving the timber self-sufficiency rate planned by the Forest Service. To create economic and environmental value from the forests, an effective harvesting plan is a good starting point. The basic purpose of forestry is sustainable timber production throughout the entire process of forestry, including planting, management, harvesting, and replanting. Sustainable timber production would lead to efficiencies of the next process, including manufacture, distribution, and sale of timber products. To examine the efficiency of the timber production sector, a guideline for optimal harvesting should be discussed first.

This study defines the condition of optimal harvesting as a harvesting plan that maximizes the forest owners' utility. The utility maximized harvesting level derived from the discrete-time utility model implies that in order to maximize the forest owners' utility, harvests should significantly increase compared with the current level. Similar results have been obtained from a 2017 study [19]. They predicted the optimal future harvest levels and total carbon sequestration in standing trees of the ROK's forests using the global timber model. Their model predicted that current harvests should be 8,200,000 $m^3$ (including coniferous forest, broadleaf forest, and mixed forest), which is more than twice the actual current harvest level of 2,600,000 $m^3$. Further harvests should be even higher as more forest areas reach the optimal rotation age. The global timber model also projected that the appropriate harvest would be expected to increase to 20,000,000 $m^3$ in 2035.

The projected future coniferous round wood demand from An et al.'s [1] Timber Demand Model in 2050 is 11,355,000 m$^3$. The available timber supply under the utility maximized scenario, 11,522,000 m$^3$, would be expected to meet the domestic round wood supply. Some experts may be concerned about market oversupply due to increasing the harvest. However, the increasing harvest may also ease concerns over securing the domestic wood supply. For the local timber industry, the main reason to not use domestic wood is the challenge of accessing sufficient supply quantities. In Korea, the total amount of logging is limited, and harvesting is allowed only in certain seasons, such as spring. Therefore, small local sawmills that manufacture timber products using domestic wood often have difficulty with the supply shortage of wood during the winter season. Even though the Timber Products Supply and Demand model suggested by An et al. [1] projected that the demand for domestic wood is expected to increase due to restrictions on timber exports from major exporting countries and the international movement to prevent illegal logging, the total demand for round wood to produce wood products is expected to decrease in the ROK due to population decline and softening market demands. However, An et al.'s [1] approach did not take into account new demand of higher wood products such as value-added wood furniture and flooring. The actual demand of round wood might be expected to be much higher than the projected result as increment of the aging, but wealthier population in the future. To meet future domestic demand growth in Korea, it is necessary to gradually increase the country's harvesting levels in accordance with an optimal harvesting strategy. This study suggests this plan as an ideal goal that should be pursued over a long period of time rather than an immediate goal for next year. This is due to the lack of harvesting infrastructure such as equipment for mechanization harvesting and negative cultural perceptions of large-volume clear cutting. In addition, government budgets to subsidize replanting should be increased to greater than the current level.

Even though the baseline scenario was developed to verify the research hypothesis, the assumption of the scenario that the current harvesting level will continue in the future seems unlikely to properly reflect the reality. From a practical point of view, as the forest matures, the level of harvesting is likely to be increased. In the ROK, various techniques such as LP and Fuzzy-LP were applied in the practice to determine optimal harvest level. To compare the performance of various models including LP, Fuzzy LP, and the discrete-time utility model in terms of wood production and consumer utility might be a substantial topic for future research.

## 5. Conclusions

In this study, we introduced the discrete-time utility model to examine an optimal harvesting decision on multiple age-class private forests that maximizes private forest owners' utility in the ROK. Our model assumed that the forest owner's decision to maximize utility determines the optimal harvest strategy through their consumption levels and the constraint is the sum of the total consumption which cannot exceed the total profit generated from forestry. Dissimilar to other studies using LP models [13], this study did not set the ending age constraint to improve the current unbalanced forest age structure in the ROK. Instead, this study showed that the unbalanced age structure might be improved through the harvest prescription derived from the utility model.

This study hypothesized that current harvesting levels of privately owned coniferous forests in the ROK are not optimized. Our study also suggested better harvest decisions by comparing two scenarios, including a baseline scenario and a utility maximized harvest scenario. Summarizing the annual wood supply can be achieved through these two scenarios, and the results estimate that over 100 years, the mean annual wood supply under the baseline scenario (status quo) will be 1,315,000 m$^3$, while the mean annual wood supply will increase to 11,522,000 m$^3$ under the utility maximized harvest scenario. These results imply that harvests should be significantly increased to maximize the forest owners' utility. To look over the age structure dynamics, as time passes, the area of younger age class decreases, and the area higher than age class 6 largely increases under the baseline scenario. Therefore, irregularities in the age distribution are expected to be kept in perpetuity. Nevertheless, the area between age classes becomes balanced, and the initially unbalanced distribution of age class is being

closed to uniform at the end of the planning horizon under the utility maximized harvest scenario. The projected demand predicted that at least 3,407,000 m³ of domestic coniferous round wood is required annually to achieve the policy goal of a 30% self-sufficiency rate in timber. The projected timber supply under the utility maximized scenario would be adequate to achieve this policy goal. However, timber supply is expected to be insufficient by 2050 under the baseline scenario. The baseline scenario is expected to meet only approximately half of the projected demand for domestic coniferous round wood. The utility maximized harvest scenario also improves the forest owners' utility level. For example, the harvest decision derived by the utility maximized scenario increases the utility level of forest owners by 15% over 10 years compared to harvesting decisions following the baseline scenario.

**Author Contributions:** Data creation and Methodology, S.L. and H.A.; Project administration, H.A.; Supervision, S.L.; Writing—original draft, H.A.; Writing—review and editing, S.L. All authors have read and agreed to the published version of the manuscript.

**Funding:** This study was supported by a grant from the Korea Rural Economics Institute (No. R876)

**Conflicts of Interest:** The study claims no conflict of interest.

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
