# Peer review of "A Utility Maximized Harvest Decision Model for Privately Owned Coniferous Forests in the Republic of Korea"

_forests, doi:10.3390/f11121273_

Round 1
Reviewer 1 Report
The manuscript: “A utility maximized harvest decision model for privately owned coniferous forests in Korea” deal with interesting problem –optimalization of harvesting decisions in management of multiple age-class forests. The novelty of research problem is to develop discrete-time utility model and use it in optimalization of harvest decisions. It could be interesting for Forests readers especially that unequal age class distribution is a problem of harvest determining not only in Korea but in other countries also. Manuscript need substantial improvement - some suggestion are described below.
General remarks:
- The assumption in basic scenario that forest in the future will be harvested at the same level as average harvest in 2017 and 2018 is not acceptable. Currently the area of stands in rotation age (45 year) class 5 is 348 *103 In the next decade stands from age class 4 (908 *103 ha) will moved to class 5 – matured for harvest and area increased more than double. So level of harvest volume increased also when calculated using Linear programming (LP), Fuzzy-LP, Geometric programming (GP) 85 and Fuzzy-GP as was stated (line 84) that this methods was used in practice in Korea. Mentioned above methods are widely used for harvest optimalization and described in the literature, so some of these articles should be cited and discussed in this manuscript.
- Some of formula presented in manuscript should be simplified - there is a lot of replication in formulas nr: 6,7 - described value of forest and nr: 9,10 used for harvest determination but base on relation of forest value in age j (left had site of formula 9 and 10) and value in age j+1 (right had site of formula). Left and right site of formulas9/10 is almost the same they differ only one index “j” and j+1. It will be more easy to understand presented relation when one use in formula 9/10 symbol of forest value determined in formula 6/7. In presented form formula 9 /10 is not equation but inequality.-
- The choice of optimal harvest level in utility maximized harvest scenario based on table 5 is not clear. According to relation described by formula 10 when value of forest in age” j+1” is smaller than in year before “j” the best choice for harvest is in year “j”, (in our case age class 4) so why (line 358) “preferential harvesting of age class 6 is a better decision in terms of profit and timber supply than simultaneous harvesting of age class 4–6”
Specific comments:
Line 47 check if “2,000 m3” is correct, probably it should be “2,000 m3 thousand”
Line 103. “Carbon is released when trees are harvested … “ not all carbon is released – in this panuscript it is assumed that 50% of carbon is released.
Line 112 Mendelsohn [9] – it should be Mendelsohn et al. [9]
Line 182 – some symbols used in formula 5 are not explain.
Line 185 formula nr. 5 is not equation. It is not “constraint” objective function which value should be maximize
Line 221 “ … the volume of per unit area …” –delete “of”
Line 315 Table 2 - Korean Won/Euro exchange rate 1,296 – is it correct – probably it shoul de 1.296
Line 375 figure 2 – why there is no age class 5 and occure class 6 in period longer than 4.
Line 344 title of table 5. “ … by Age class” it should be “ age class”
Author Response
The attached file includes the author's notes to reviewer

Reviewer 2 Report
Adopting an appropriate forest use strategy is a very important issue not only from an economic but also a social point of view. The demand for wood raw material in the world is constantly increasing. Countries have different timber supply strategies. Some of them, at the lack of domestic raw material, simply import it. Others, through their decisions making, try to supply it from their own resources. One of the possibilities in this regard is production optimization. The authors of the article took up the topic related to making decision of harvest prescription derived from the discrete-time utility model with a multiple age-class forest. It is valuable as it has been prepared by people professionally involved in economic issues.
However, accepting the article for publication requires some corrections. First, the Authors should make it clear which Korea it concerns. The affiliation shows that it is the southern part of Korea. Therefore, they should use the name of „Republic of Korea”.
Introduction should be extended to the available world literature on different approaches to the supply of wood from domestic markets. On of the examples would be the article "Designing Wood Supply Scenarios from Forest Inventories with Stratified Predictions", published in Forests.
In Introduction, subsection 1.1 was introduced. This is an error, because subsections are not numbered if there is only one. There should be placed a paragraph at the end of this chapter that describes the purpose of the work.
The reviewer has no major comments on the methodological approach and the results achieved. It shows great professionalism and continuity of the main idea.
However, changes should be made in the discussion. In its present form, it is a kind of continuation of the results. The discussion should be written in a more general way and authors should relate to how the results of this work compare with the achievements of other authors. Tables and figures should be avoided in the discussion. If possible, they should be moved to the previous chapter.
Conclusions also need improvement. In this section, the Authors should refer directly to the results achieved, avoiding general considerations.
Detailed comments:
- The article should be written work impersonally, avoiding expressions such as "we".
- Lines 19, 47 and others. The authors have a problem with units. It is not possible for Korea to harvest 1.3 thousand m3 of soft wood annually. It should be a unit 1000 times larger. Please review the entire article carefully and correct the appropriate numbers.
- The first three sentences of the Introduction should be moved at the end.
- What is the share of private forests in Korea?
- Lines 62-75. This part of the text should not appear in Introducion. It should be moved to the methodology.
- Delete please Section 2.2 and move the text to the Introduction.
- Lines 188-203. Proceed with the text similar to the above.
- Line 289. Footnote3 – there is no spaces before 3.
- Table 4. There are not spaces before the brackets.
- At the end of the description of tables, in some cases are missing dots. In addition, the units should be checked throughout the work. Sometimes cubic meters were misspelled. See description below in table 7.
- Table 8 is too small. It should be deleted and replaced with text.
- The numbers should be standardized. Sometimes there is e.g. 1,000 thousand and sometimes 1,000,000.
- Line 484. Delete the Patents section.
- As mentioned earlier, the available literature should be significantly expanded.
- References should be prepared in accordance with the journal's requirements. In some places, dots are missing at the end of individual literature items.
Author Response
The attachment includes the author's notes to the reviewer

Round 2
Reviewer 1 Report
The second version of manuscript: “The utility maximized harvest decision for privately owned coniferous forest in Korea” in my opinion has been significantly improved and can be published in Forests.
Author Response
Thanks for the comments. Your comments have been of great help in the progress of our manuscript.
Reviewer 2 Report
Please remove 1.1. Research background because there is only one sub-chapter.
In Figures by descriptions of y lines, there is no space between numbers and units.
After corrections, the article can be published.
Author Response
Thanks for your comments. We remove the title of sub-chapter in the introduction. We also add some space between numbers and units in all graphs.
Your comments have been of great help in the progress of our manuscript.
This manuscript is a resubmission of an earlier submission. The following is a list of the peer review reports and author responses from that submission.
Round 1
Reviewer 1 Report
There were no line numbers in the manuscript. All comments and edits are in the .pdf.
The equations need some improvement in terms of explaining them. There was some confusion on the variables used in them. Explanation of these variables will help. Why was afforestation considered for existing forest on the one equation? Age classes need to be defined to what tree ages they represent.
There were many instances of grammar and sentence structure concerns throughout the entire manuscript. This will help the flow of the manuscript. The manuscript was also lengthy.

Author Response
Thanks for your comments. The attachment includes the author's reply.

Reviewer 2 Report
The work concerns a very interesting topic, which can be understood not only locally but also globally.
Unfortunately, the chapters: Abstract, Introduction and discussion must be completely redesigned.
Abstract requires a revision to improve the comprehension of the work.
The introduction is deprived of an analysis of the current literature of the examined subject. I do not see in it the main stream which would end in the indication of the research hypothesis and the clearly demonstrated purpose of the research.
I found also some inaccuracies in formulas No. 3,6,7. But maybe these are only editing errors.
line 9 Abstract should be rewritten - following the instructions in the Forests articles.
line 20 Citation in the abstract ... Not acceptable in my opinion. As the authors justify the need to use this particular citation? If so, then Forests has different citation rules!
line 20 There is no space between the cited authors name and the brackets.
line 31 Introduction should be rewritten! This work is supposed to maximize harvest decision - and initially there is not even one information on how much timber is currently obtained (tones, m3). The lack of references to current research in the world with particular emphasis on the precursors of this research in Europe and America (except for the spontaneous use of Finnish publication in the abstract) means that this chapter cannot be accepted.
line 69 no spaces between citation and brackets.
line 75 no spaces between citation and brackets.
line 95 no spaces between citation and brackets.
line 96 no spaces between citation and brackets.
line 105 no spaces between citation and brackets.
line 142 no spaces between m3 and of timber
line 152-153 error in pattern transformation :
from (1) - A-D=
A-delta(1-delta)^(k-1)*A=
A[1-delta(1-delta)^(k-1)]<>
(1-delta)^k*A - from (3)
<> - differ
e.g delta=1/2, k=3
(1) A*[1-1/3(1-1/3)^(2)]=A*[1-1/3(2/3)^2]=A*[1-1/3*4/9]=A*[1-4/27]=A*[23/27]
(3) A*{1-1/3]^3=A*(2/3)^3=A*[8/27]
Line 206 indexing: first x is with the index 0 at the top (page 6, 3 row from the top) in formula (6) is the index and also at the top, in formula (7) the index at x is at the bottom and a second index appears: 0i so maybe this is a different x? this is not explained.
line 297 you start by describing the results of your own work by presenting the results of other research. It looks a bit like a student's mistake when writing the first version of the master's or bachelor's thesis.
line 318 graph must be changed - in this version is unreadable.
Line 394 Discussion should be rewritten Sorry, but once again - the same problem appears as in the introduction. This is not a properly written discussion. It should include discussing the results not only with the views of the autoar, but above all with the results of related research available.
Line 469 Testifies that the authors did not thoroughly examine the available literature on the subject

Author Response

(The authors gave the same response as above.)

Reviewer 3 Report
This manuscript presents a set of financial equations to calculate the optimal rotations for existing forest and regenerated forest in Korea. It implements some of those equations to support a large scale forest planning (I suppose it is for the entire Korea) to calculate the desired national harvesting strategy and harvesting level.
I think this manuscript covers an important subject but I have three major concerns about this manuscript: 1) how realistic is it to generalize the conifer forest in Korea only into several age classes? Would this research be more meaningful by further classifying the forest into at least cover types, site quality, and harvesting accessibility? 2) This manuscript needs substantial English editing. There are many grammar issues. Notations are not used consistently across the paper. The writing is sometimes confusing. 3) I cannot find clear explanation about how forest level concerns such as even flow of timber production is achieved.
Here are just a few examples regarding poor English only from the first two paragraphs. I don’t have time to list all the problems. The authors need to do a better job in proof reading.
- “Currently, Korea has 6.19 million ha are stocked with trees.” This is not a correct sentence.
- “by products” should be “byproducts”.
- “This makes forest owners difficult to earn profit from planting and selling trees.” Should be “This makes it difficult for forest owners to earn profit from planting and selling trees.”
- “has proposed many programs to consistent with” should be “has proposed many programs to be consistent with”
- Not sure what does “the cycle changes in timber production level” means?
- “This might be hinder stability of timber supply”. Hinder is a verb.
Based on Equation 4, natural regeneration is not considered after a forest died. It seems all the new forests must be planted. Is this true in Korea?
In equation (3), the authors need to state that the unit of t, i and k are all years; otherwise, if t is measured by decades, t-k would not make any sense.
The notations are not consistent and confusing, i.e. δ is used in both equation 4 and 5, but to denote different parameters. There are same problems for the use of k between equations 4 and 5.
“We assume that CO2 sequestered in the tree was released about 50%...” Any citations, or is this number just made up?
Before equation (8), “ct” should be “c subscript t.
“If logging is decided for the existing forests and then harvest plan for the bare-land is applied, equilibrium can be found regardless of the type of the existing forest.” I don't understand this statement. I know each forest type may have an optimal rotation, but we also need to coordinate the harvesting of different forest types to achieve forest level objectives such as even flow of timber. The main objective of forest level (in contrast to stand level) optimization is to achieve forest level objectives.
“For the analysis, the above model was applied to a privately owned coniferous forest in Korea …” Is this test case built for the entire Korea or just for a private forest? confusing to me.
Although figure 3 shows quite consistent timber harvesting levels across the next 180 years, this paper does not explain how this even flow of timber harvesting is achieved. The authors mentioned “Therefore, we conclude that the best strategy is to prior harvesting of age class 5 and 6 in the beginning of the planning period, then start the harvesting of age class 4 from the 4th period.” Do you mean you will follow the oldest first rule to harvest all forest in age class 5 and 6 at decade one? This is not stated clearly in the paper.
Author Response

(The authors gave the same response as above.)

Round 2
Reviewer 2 Report
Unfortunately, the sections Introduction and Discussion have not been corrected as suggested. The authors did not make any effort to study literature from other countries (apart from a few publications). This is evidenced by, for example, a slightly enlarged literature list. As I suggested the work is very interesting, however, it takes months, not weeks, to reconstruct it and make it acceptable. Considering the current state of the work, I suggest you reject it. However, I encourage the authors to resume their efforts and submit the work once again, however, in a form that meets the current requirements of the research work.
Author Response
The attachment includes the author's answer for the comments

Reviewer 3 Report
I listed three major concerns in my review. I don't feel the authors addressed my concerns sufficiently.
1) how realistic is it to generalize the conifer forest in Korea only into several age classes? Would this research be more meaningful by further classifying the forest into at least cover types, site quality, and harvesting accessibility?
No response from the authors. Should site index at least be included in the analysis as a timber harvesting paper?
2) This manuscript needs substantial English editing. There are many grammar issues. Notations are not used consistently across the paper. The writing is sometimes confusing.
Their response is: "For English writing, we plan to ask for editorial service provided by forests, and we expect that grammatical error will be corrected through their proof reading."
This is the first time I got such an answer. Should the authors fix the language problem first before resubmitting the manuscript for further review? I am confused by the process.
3) I cannot find clear explanation about how forest level concerns such as even flow of timber production is achieved.
Their answer is: In this strategy, age class 6 would be harvested first, but the amount of harvesting volume of age class 6 cannot satisfy the amount of consumption (12,000 10 billion over 10 year), the following lower grades such as age class 4 and 5 are assumed to be harvested.
Here they explained the harvesting sequence, basically older forest first, but they failed to explain why the 12,000 10 billion over 10 year is an appropriate and sustainable harvesting target. We cannot simply calculate the sustainable harvesting level based on "consumption." We typically have to use area control, volume control, simulation or optimization models to test and select the sustainable harvesting level, especially for a long term plan.
Author Response

(The authors gave the same response as above.)
